# Grounding by Trying: LLMs with Reinforcement Learning-Enhanced Retrieval

**Sheryl Hsu[1], Omar Khattab[1,2], Chelsea Finn[1,3] & Archit Sharma[1,4]**
[1]Stanford University,[2]Databricks,[3]Physical Intelligence,[4]Google DeepMind
{sherylh,architsh}@stanford.edu

## Abstract

The hallucinations of large language models (LLMs) are increasingly mitigated by allowing LLMs to search for information and to ground their answers in real sources. Unfortunately, LLMs often struggle with posing the right search queries, especially when dealing with complex or otherwise indirect topics. Observing that LLMs can learn to search for relevant facts by *trying* different queries and learning to up-weight queries that successfully produce relevant results, we introduce Learning to Retrieve by Trying (LeReT), a reinforcement learning framework that explores search queries and uses preference-based optimization to improve their quality. LeReT can improve the absolute retrieval accuracy by up to 29% and the downstream generator evaluations by 17%. The simplicity and flexibility of LeReT allows it to be applied to arbitrary off-the-shelf retrievers and makes it a promising technique for improving general LLM pipelines. Project website: http://sherylhsu.com/LeReT/.

## 1 Introduction

Despite tremendous progress, large language models (LLMs) still often hallucinate, motivating significant interest in grounding LLM answers in verified sources (Guu et al., 2020; Komeili et al., 2022; PerplexityAI, 2024; Google, 2024; OpenAI, 2024). Unfortunately, simply retrieving semantically similar documents to the user question, as is prevalent in retrieval-augmented generation (RAG; Lewis et al. 2020) pipelines, tends to fail for complex information needs not answered directly by any individual document. To tackle this, *multi-hop retrieval* pipelines gather information incrementally over multiple steps of search. For example, if a user asks *What is a good dinner place driving from the Bay Area to Lake Tahoe on Friday night to avoid traffic?*, a grounded system might need to learn about `towns en route Lake Tahoe from the Bay Area`, followed by `traffic forecast on I-80` and finally, `restaurants in Auburn` (and other towns).

However, doing this successfully is hard as off-the-shelf LLM performance is often unsatisfactory, and producing supervision for the best search queries to generate in a sequence of "hops" is non-trivial and expensive. Recent work tackles this via *prompt optimization* and *rejection fine-tuning* given a downstream signal. For example, Khattab et al. (2023) "bootstrap" trajectories of reasoning and search queries and collect trajectories that lead to a high downstream answer accuracy, using them either to search for effective few-shot prompting examples or to finetune the LLM responsible for query generation. We observe that the problem of teaching a LLM to generate effective search queries is inherently a reinforcement learning (RL) problem and ask *can RL improve the grounding of answers generated by LLMs when wielding open-ended tools like search engines?*

If we can observe the retrieved documents for different search queries and compute rewards for finding relevant documents, we can train the LLM to produce queries that lead to better outcomes. Such learning from trial-and-error naturally lends itself to RL formalism, going beyond imitation-based methods in prior works. Indeed, we find that naïve sampling from LLMs with high temperature and using the observed data for RL is not effective. Instead, our proposed framework, *Learning to Retrieve by Trying*, or LeReT, induces diverse search queries for each question by diversifying the few-shot examples in the prompts of the system. After this diversified sampling of search queries and the resulting retrieval, LeReT applies context distillation (Snell et al., 2022) followed by opti-

Performance of Different Algorithms for Grounding LLMs

Figure 1: **LeReT significantly improves retrieval and generation.** LeReT provides a reinforcement learning based framework for improving grounding and performance of LLM generated answers by improving the retrieval of relevant factual data.

mizing the query-generating LLM using preference-based RL. We use identity policy optimization (IPO; Azar et al. 2023; Rafailov et al. 2024b), though other RL algorithms can be substituted.

Our main contribution is LeReT, a framework for improving grounding of LLM answers by leveraging retrieval annotations to improve multi-hop retrieval accuracy. On two question-answering datasets, LeReT considerably outperforms prior methods like few-shot prompting and supervised fine-tuning in both retrieval quality and downstream generation quality, with stronger generators like GPT-4 benefiting more from the improvements in retrieval. We experiment with an iterative version of LeReT and find that its performance improves over iterations. Our analysis reveals that prompt-driven diverse sampling is critical for LeReT to be effective, and we also analyze different ways to generate rewards for retrievals. Finally, our experiments find that LeReT can be used across retrievers, and thus, provides a simple and general framework for improving retrieval. While we focus on retrieval for grounding LLM answers in this work, the core method behind LeReT can be straightforwardly extended to other agentic pipelines in which LLMs control a blackbox tool, so long as a reward can be formulated on its outputs.

## 2 RELATED WORK

**Retrieval-Augmented Generation (RAG).** Over the past few years, interest has been growing in conditioning LLM outputs on retrieved information (Chen et al., 2017; Lee et al., 2019; Guu et al., 2020; Lewis et al., 2020; Lazaridou et al., 2022; Asai et al., 2024). This strategy seeks to make LLM systems more efficient, updatable, and transparent by decoupling the system's knowledge from the model parameters. This makes it easy to update the knowledge corpus and also makes it possible to inspect the sources relied upon when LLMs produce factual statements.

Previously, Nogueira & Cho (2017) trained a query reformulator for retrieval queries using reinforcement learning. While this is the closest analogue to our work, this paper does not use a generative language model but instead presents a very specific query selection architecture (understandably, since it is 2017) and corresponding training recipe. A straightforward application of RL similar to this applied to modern LLMs would result in performance similar to few-shot prompting.

**Multi-Hop Retrieval.** The standard RAG formulation is best suited for "simple" questions, where a direct search can find all the information required for producing responses. Beyond these, benchmarks such as HotPotQA (Yang et al., 2018), MuSiQue (Trivedi et al., 2022), and HoVer (Jiang et al., 2020) assess systems on gathering and synthesizing information from several independent documents within a massive corpus like Wikipedia. To tackle retrieval in this setting, early systems like MDR (Xiong et al.) and Baleen (Khattab et al., 2021) introduced bespoke strategies for fine-

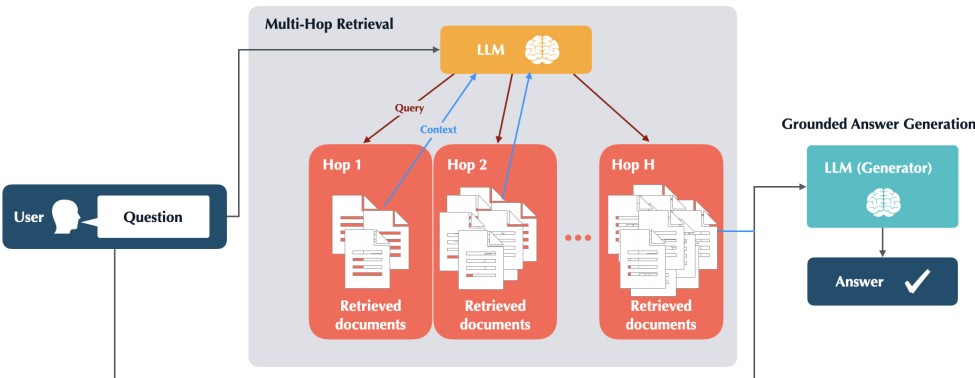

Figure 2: An overview of the standard multi-hop retrieval pipeline we study in this work. A user asks a question to the system. In each hop, the LLM generates search queries for the retriever and receives a collection of documents. The overall set of retrieved documents and the user question are then given to a downstream LLM for grounded answer generation.

tuning the retrieval models that produce representations of queries and documents, adapting them directly for compositional search queries. Unfortunately, fine-tuning the retriever representations is a data hungry approach that is also challenging to scale to massive corpora like the Web, as re-training the retriever often requires re-indexing the corpus. Increasingly, research in this space (Trivedi et al., 2023; Yao et al., 2023; Khattab et al., 2022; Press et al., 2023) relies on off-the-shelf retrievers such as the Wikipedia API and focuses on improving LLMs' ability to to generate effective queries.

**Optimizing LLM Programs & Agents.** Recent work tackles this by employing *prompt optimization* and *rejection fine-tuning* using a downstream signal. For example, in Khattab et al. (2023), the authors "bootstrap" trajectories of reasoning and search queries and use the trajectories that lead to a high downstream answer accuracy as candidate examples. The trajectories are then used either to search for effective few-shot prompting examples or to finetune the LLM responsible for query generation. This approach was extended in Opsahl-Ong et al. (2024) and Soylu et al. (2024) in which the authors also explore using these trajectories to search for free-form instructions for prompting or to nest forms of rejection fine-tuning and prompt optimization, respectively.

Additionally, prompt-based techniques have been developed to further improve retrieval and downstream generations. For example, Query2doc (Wang et al., 2023) prompts the LLM for hypothetical documents to concatenate with the query. Rethinking with Retrieval (He et al., 2022) uses decomposed reasoning steps as queries. On the downstream generation side, there have been new methods (Yu et al., 2024; Lan et al., 2023; Kim et al., 2023) that enable the generator to better reference the retrieved information and produce better answers. As a general RL framework, our work is complementary to these prior works.

Beyond retrieval or LLM programs, similar techniques can be used for optimizing agent behavior. For example, in Song et al. (2024), the authors train agents to navigate the web, simulate science experiments, or perform household tasks by collecting failure and success trajectories and training the LLM using preference optimization. Our work finds that the quality of exploration data sampled from LLMs is critical to the success of RL in agentic pipelines, and introduces a prompt-based diversification strategy that samples diverse and high-quality exploration data (discussed in Section 4.1, analysis in Section 5.4).

## 3 PRELIMINARIES

**Multi-Hop Retrieval Setup**. An overview of retrieval is shown in Figure 2. We assume access to a retriever that maps a search query $q$ to a set of $N$ most similar documents $D = \{d_i\}_{i=1}^{N}$, where $d_i$ denotes an individual document. A user asks a question $u$, a LLM $\pi_r$ generates a query $q_1$ for the retriever, which results in an ordered set of document $D_1$. In the next hop, the LLM $\pi_r$ takes $u$ and $D_1$ as input and outputs query $q_2$. This repeats for $H$ hops. The final ordered set of retrieved documents $D_R = D_1 \cup D_2 \cup \ldots D_H$ is given as input to the LLM generator $\pi_g$ along with the

question $u$, which generates the final answer for the user query. In this work, we restrict ourselves to fine-tuning the LLM $\pi_r$ and treat the retriever and LLM generator $\pi_g$ as blackbox models.

**Language Models and Reinforcement Learning**. Reinforcement learning has become the de facto tool for aligning large language models and has inspired considerable work at the intersection of language models and RL (Stiennon et al., 2020; Ouyang et al., 2022; Zhao et al., 2023; Rafailov et al., 2024b). We briefly review direct alignment methods (Rafailov et al., 2024b). Given a dataset of preferences $\mathcal{D}_p = \{x^i, y_w^i, y_l^i\}$, where $x^i$ denotes the dialogue history, $y_w^i$ denotes the preferred response, and $y_l^i$ denotes the dispreferred response. Bradley-Terry (Bradley & Terry, 1952) offers a model that connects choice to an implicit goodness score, useful for learning a reward model $r_\phi$:

$$\mathcal{L}_{\text{BT}} = -\mathbb{E}_{(x,y_w,y_l)\sim\mathcal{D}_p}\left[\log p_\phi(y_w \succ y_l)\right] = -\mathbb{E}_{(x,y_w,y_l)\sim\mathcal{D}_p}\left[\log \sigma\left(r_\phi(x,y_w) - r_\phi(x,y_l)\right)\right], \quad (1)$$

where $\sigma$ denotes the sigmoid function and $p_\phi(y_w \succ y_l)$ denotes the probability of $y_w$ being preferred over $y_l$. Typically a LLM $\pi$ is trained to maximize this learned reward model using RL as described by $\max_\pi \mathbb{E}_{y\sim\pi(\cdot|x)}\left[r_\phi(x,y) - \beta\text{KL}(\pi \| \pi_{\text{ref}})\right]$, where $\pi_{\text{ref}}$ denotes a fixed reference policy. However, Rafailov et al. (2024b) shows that parameterizing $r_\phi(x,y) = \beta \log\left(\pi_\phi(y \mid x)/\pi_{\text{ref}}(y \mid x)\right)$ and optimizing Eq 1 implicity optimizes the RLHF objective exactly, removing the need for a separately parameterized reward model or an explicit RL training loop. However, Azar et al. (2023) and Rafailov et al. (2024a) have found that optimizing a DPO parameterized reward uing Eq 1 can lead to overoptimization, and suggest optimizing the following objective:

$$\mathcal{L}_{\text{IPO}} = \mathbb{E}_{(x,y_w,y_l)\sim\mathcal{D}_p}\left[\left(\tilde{r}_\phi(x,y_w) - \tilde{r}_\phi(x,y_l) - 0.5\tau^{-1}\right)^2\right] \quad (2)$$

where $\tau$ is a hyperparameter controlling the target margin between the implicit rewards for $y_w$ and $y_l$, and $\tilde{r}_\phi(x,y) = \log\left(\pi_\phi(y \mid x)/\pi_{\text{ref}}(y \mid x)\right)$. Minimizing the objective in Eq 2 leads to identity policy optimization (IPO; Azar et al. 2023).

## 4  LeReT: Learning to Retrieve by Trying

We introduce Learning to Retrieve by Trying, or LeReT, a novel framework for improving the grounding of LLM generations by training the search query LLM $\pi_r$ using preference optimization. In hop $i$, we sample a query $q_i$ from the LLM based on the user question and documents seen in hops $< i$, and observe a reward signal for the retrieval quality. Both sampling from the LLM and retrieval make it hard to backpropagate directly from the reward signal, making RL a more suitable optimization framework for this setting. We first discuss how to generate a dataset of queries and retrieved documents that is suitable for RL optimization in Section 4.1. We then discuss how to convert the reward-annotated dataset into a dataset of preferred and dispreferred queries and use that dataset to optimize $\pi_r$ with IPO in Section 4.2. We also briefly discuss an iterative version of LeReT that alternates between sampling and optimization. Finally, we combine the elements and give a practical overview in Section 4.3.

### 4.1  Prompt Driven Diverse Query Generation

Given a dataset of questions, we want to "try" a set of search queries and observe the retrieved documents. What queries would be good to observe the retrieved documents for? This roughly corresponds to the exploration problem in RL. As our experiments later in Section 5.4 also indicate, a good distribution of queries would result in diverse outcomes (for better exploration), but it is important that some queries produce high quality retrievals. To sample such diverse and effective queries, LeReT moves beyond high-temperature sampling and uses a diverse set of examples to few-shot prompt the LLM $\pi_r$. We use DSPy (Khattab et al., 2022; 2023)'s prompt optimizers, specifically a simple `BootstrapFewShotWithRandomSearch` (BFRS), to self-generate a number of independently optimized few-shot, chain-of-thought prompts $\mathcal{P} = \{p_1, \ldots, p_P\}$ for LLM $\pi_r$. The independently optimized prompts would naturally lead to diverse samples from $\pi_r$, and DSPy's optimization ensures that the prompts are still resulting in relevant retrievals. Note that we can reuse the same set of prompts across all questions throughout the dataset.

For a hop $h$, LeReT does the following: For every question $u \in \mathcal{U}$, LeReT samples search queries conditioned on each of the prompts $p \in \mathcal{P}$, resulting in a set of search queries $Q_h = \{\pi_r(\cdot \mid p_i, u, C_{h-1})) \mid p_i \in \mathcal{P}\}$, where $C_{h-1}$ denotes the context from previous hops. For

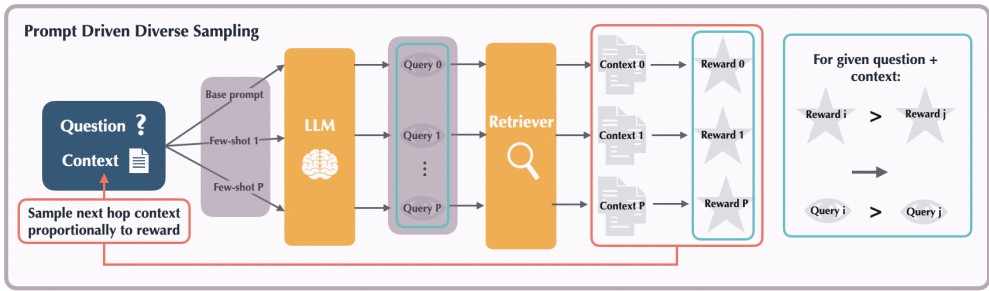

Figure 3: **Overview of prompt driven diverse sampling and data generation.** LeReT induces diverse but effective search queries by bootstrapping several few-shot prompts for query generation and uses the retrieval reward to collect preferred and dispreferred queries for each question's hop.

every query $q_i \in Q_h$, we retrieve a set of documents denoted $C_{hi}$ and compute the reward corresponding to each query by evaluating the retrieved documents, that is $r_{\text{ret}}(u, C_{h-1}, q_i) = \mathbb{R}(C_{hi})$ (more on retrieval reward $\mathbb{R}$ in Section 4.3). The final dataset for this hop consists of $\mathcal{D}_h = \{(u, C_{h-1}, q_i, r(u, C_{h-1}, q_i)) \mid u \in \mathcal{U}, q_i \in Q_h\}$, where every entry consists of the user question, the context from the previous hop, the sampled query, and the reward computed by running the retriever. Though the prompts we generate are used to sample diverse high quality search queries at training time, we leverage context distillation to remove the need for optimized prompting at test-time.

When training in a multi-hop setting like this, we must select which contexts to use for the next hop. Naïvely generating queries for every possible context leads to an exponentially growing dataset with respect to the number of hops, which becomes computationally infeasible quickly. At the end of hop $h$, we have $P$ different contexts $C_{hi}$ for a given question. LeReT randomly selects one of the $P$ contexts to use for the next hop, creating $C_h$. To do this, we first filter out contexts that have achieved the optimal reward (assumed to be 1), as no more relevant documents can be retrieved. For the remaining contexts, we weigh each context by the reward and sample one of them. This biases the data towards trajectories that achieve higher reward, while still containing trajectories where the model can recover from poor retrieval in earlier hops. The final training dataset is simply the union of the dataset from each hop, that is $\mathcal{D}_{\text{tr}} = \mathcal{D}_1 \cup \ldots \mathcal{D}_H$. The pseudocode for dataset sampling is given in Algorithm 1.

## 4.2 Model Optimization

Given the training dataset $\mathcal{D}_{\text{tr}}$, we want to update the LLM $\pi_r$. First, we make a simplification by optimizing every hop greedily, that is, based on the reward obtained in that hop alone. Ideally, updating $\pi_r$ on data in hop $h$ accounts for rewards obtained in all future hops. Retrieving relevant documents in earlier hops is likely always better, and intuitively, retrieving irrelevant documents in earlier hops will rarely lead to better overall retrieval. We verify this assumption empirically: For two sets of retrieved documents, a low reward and high reward set, a low reward retrieval leads to a higher total reward in only 0.026% of the cases (Appendix B.1). Thus, the greedy approach considerably simplifies the optimization, without any evident theoretical or empirical sacrifice.

Given the dataset $\mathcal{D}_{\text{tr}}$, we can optimize $\pi_r$ using any RL algorithm. In this work, we consider preference-based RL approaches, specifically IPO described in Section 3 because of its simplicity and effectiveness. To optimize $\pi_r$ using IPO, we need to transform $\mathcal{D}_{\text{tr}}$ into a preference dataset. This can be done straightforwardly by comparing two search queries $q_i$ and $q_j$ for the same user question and context and choosing the query with higher reward as the preferred response and the other query as the dispreferred response. However, before we can optimize $\pi_r$ using IPO, we must account for the fact that the search queries were sampled using few-shot prompting, and at test-time we will not be using the prompt. To do so, we leverage context distillation (Snell et al., 2022) by fine-tuning on the search queries without the context. We observe in our experiments that this roughly matches the performance of few-shot prompting. After context distillation and converting the training dataset into a preference dataset, we optimize $\pi_r$ using IPO.

**Iterative-LeReT**. Thus far, we have assumed that sampling search queries and model training are done as two separate steps. However, we can alternate between the two steps, leveraging the improvements in previous iterations to sample better data for the next iterations. Iterative-LeReT closely follows iterative-DPO (Xu et al., 2024). We partition the dataset of user questions $\mathcal{U}$ and run LeReT on each partition, sampling from the model fine-tuned on the previous partitions. Specifically, we have $I$ data partitions $\mathcal{U}_1, \ldots, \mathcal{U}_I$. We start with the LLM $\pi_0$ and apply LeReT on $\mathcal{U}_1$ to obtain the fine-tuned LLM $\pi_1$, so we have LeReT $(\pi_0, \mathcal{U}_1) \rightarrow \pi_1$. We use LeReT on $\mathcal{U}_2$, using $\pi_1$ to sample the training data and continue fine-tuning from, that is, LeReT $(\pi_1, \mathcal{U}_2) \rightarrow \pi_2$. We can repeat this for all $I$ partitions until we get the final model $\pi_I$. We find that iterative sampling and training can be effective as models may not achieve accurate and relevant retrievals in the initial iterations, and later models may be able to generate better exploration data.

### 4.3 Reward Labeling for Retrieved Documents

In order to construct the required preference datasets, we need a reward signal $\mathbb{R}$ to score the documents retrieved by a search query. How do we compute this reward signal? There are broadly two ways to get such supervision: *direct supervision*, where a human provides oracle documents to ground the answers in the training dataset or explicitly reviews the relevance of documents retrieved by the search query and *indirect supervision*, where the supervision comes from evaluations of the downstream generator such as preference feedback on the final answers or some answer verification. The latter is indirect because we do not obtain any explicit supervision for retrieval, but only receive information about how the generator performed after being conditioned on the retrieved documents. We run a short study in Section B.5 comparing the two forms of supervision, and find that direct supervision results in better performing models. However, a full study comparing direct and indirect forms of supervision and their trade-offs is beyond the scope of this paper, and potentially requires novel algorithmic considerations. For the majority of the paper, we assume some form of direct supervision, as allowed by commonly used datasets and benchmarks.

---

**Algorithm 1** Prompt Driven Diverse Sampling + Training

1: **Input:** Number of hops $H$, Number of few-shot prompts $P$, LLM $\pi_r$, Retriever $\mathcal{K}$, Dataset $\mathcal{U}$
2: **Initialize:** $C_1 = \varnothing$; $[p_1, \ldots, p_P]$ as few-shot prompts; $\mathcal{D}_{\text{pref}} = \varnothing$
3: **for** $h$ in range$(1, H)$ **do**
4:     **for** $u$ in $\mathcal{U}$ **do**
5:         **for** $i$ in range$(0, P)$ **do**
6:             Sample $q_i \sim \pi_r(\cdot \mid u, p_i, C_{h-1})$
7:             Retrieve $C_{hi} \leftarrow C_{h-1} \cup \mathcal{K}(q_i)$
8:             Compute reward $r_i = \mathbb{R}(C_{hi})$
9:         **end for**
10:        **for** $i$ in range$(0, P)$ **do**
11:            **for** $j$ in range$(i + 1, P)$ **do**
12:                **If** $r_i \neq r_j$: Add $(u, C_{h-1}, q_i, q_j)$ to preference dataset $\mathcal{D}_{\text{pref}}$
13:            **end for**
14:        **end for**
15:        Sample context for next hop $C_h \sim \text{Sample}(C_{hi}, \mathbb{P}(C_{hi}) \propto r_i)$
16:     **end for**
17: **end for**
18: $\pi_{\text{LeReT-CD}} = \text{SFT}(\pi_r, D_{\text{pref}})$
19: $\pi_{\text{LeReT}} = \text{IPO}(\pi_{\text{LeReT-CD}}, D_{\text{pref}})$

---

## 5 Experimental Evaluation

We now evaluate how LeReT impacts the quality of retrieval and of downstream generation. We first test LeReT on two multi-hop question answering datasets, finding that LeReT significantly outperforms baselines such as few-shot prompting and supervised fine-tuning. We also find that applying LeReT iteratively leads to further improvement over iterations. We analyze prompt driven diverse sampling in contrast with sampling using high temperature and also discuss different reward functions for the retrieval step. Finally, we evaluate LeReT's adaptability for various pipelines by testing it against retrievers.

| Model Dataset | Method | 1 Hop | | 2 Hops | | 3 Hops | | 4 Hops | | Generator |
| | | RE | AP | RE | AP | RE | AP | RE | AP | EM |
|---|---|---|---|---|---|---|---|---|---|---|
| Llama 8b HotpotQA | Base | 42.3 | 38.8 | 54.7 | 41.9 | — | | | | 41.0 |
| | Few-shot | 49.9 | 45.6 | 64.8 | 53.9 | — | | | | 47.1 |
| | Few-shot all | 50.2 | 46.4 | 63.5 | 50.5 | — | | | | 45.2 |
| | Query2Doc | 47.7 | 43.7 | 61.4 | 51.1 | — | | | | 44.8 |
| | LeReT-CD | 51.4 | 47.3 | 69.8 | 58.0 | — | | | | 49.3 |
| | LeReT | 56.7 | 52.5 | **77.1** | 66.3 | — | | | | **52.5** |
| Gemma 9b HotpotQA | Base | 52.2 | 48.4 | 70.9 | 57.7 | — | | | | 51.0 |
| | Few-shot | 54.4 | 50.4 | 66.7 | 57.8 | — | | | | 48.5 |
| | Few-shot all | 54.6 | 50.5 | 69.6 | 58.8 | — | | | | 50.0 |
| | Query2Doc | 43.5 | 40.0 | 51.2 | 43.9 | — | | | | 35.9 |
| | LeReT-CD | 53.5 | 49.6 | 71.9 | 59.2 | — | | | | 51.4 |
| | LeReT | 56.1 | 52.2 | **79.9** | 67.0 | — | | | | **54.3** |
| Llama 8b HoVer | Base | 37.9 | 34.8 | 45.6 | 37.9 | 48.5 | 38.3 | 50.0 | 39.3 | 61.5 |
| | Few-shot | 45.6 | 42.2 | 53.4 | 45.9 | 56.0 | 46.0 | 57.3 | 46.1 | 64.6 |
| | Few-shot all | 38.8 | 35.8 | 51.9 | 44.4 | 57.5 | 46.3 | 59.7 | 45.9 | 64.7 |
| | Query2Doc | 39.7 | 36.4 | 48.9 | 42.1 | 53.9 | 44.3 | 57.1 | 45.7 | 64.0 |
| | LeReT-CD | 42.9 | 39.8 | 56.6 | 48.4 | 63.2 | 52.2 | 66.9 | 54.3 | 67.5 |
| | LeReT | 45.8 | 42.5 | 65.4 | 56.1 | 72.8 | 61.4 | **76.9** | 64.3 | **69.8** |
| Gemma 9b HoVer | Base | 40.8 | 37.7 | 45.5 | 38.1 | 48.8 | 39.6 | 50.1 | 40.4 | 61.7 |
| | Few-shot | 46.3 | 42.9 | 55.4 | 46.8 | 57.9 | 48.4 | 59.3 | 48.5 | 64.3 |
| | Few-shot all | 46.3 | 42.8 | 55.8 | 48.7 | 64.1 | 52.6 | 68.2 | 54.1 | 67.5 |
| | Query2Doc | 40.5 | 37.6 | 43.3 | 39.1 | 45.5 | 40.2 | 46.7 | 40.7 | 60.3 |
| | LeReT-CD | 45.2 | 41.7 | 59.5 | 50.7 | 65.3 | 54.1 | 69.0 | 56.2 | 67.2 |
| | LeReT | 47.0 | 43.7 | 67.5 | 57.6 | 75.2 | 63.1 | **79.4** | 66.1 | **71.5** |

Table 1: **LeReT improves the performance of Llama 3 8b and Gemma 9b on HotpotQA and HoVer**. We compare models trained with LeReT versus the base model, the base model with few-shot prompting, and the base model with Query2Doc. We measure the recall (RE) and average precision (AP) of the retrieved documents (higher is better) along with the exact match of generations produced using the retrieved documents.

**Datasets.** We test LeReT on HotpotQA (Yang et al., 2018) and HoVer (Jiang et al., 2020). Both datasets are based on a Wikipedia knowledge base and are multi-hop, meaning that models must reason across multiple articles to arrive at the correct answer. The datasets provide both the correct answer and supporting articles. HotpotQA is a question-answering dataset that requires up to 2 hops, while HoVer is a fact verification dataset that requires up to 4 hops. Both datasets are relatively large, allowing us to train on over 10,000 questions.

**Evaluation metrics.** We measure retrieval performance using recall and average precision. Recall is the number of correctly retrieved documents over the total number of correct documents. Average precision (Eq. 3 in the appendix) takes into account the ordering of the documents, i.e. if the correct 3 documents are ranked as the last 3 out of 6 the score will be lower. For generation, we measure both exact match on the entire answer and F1 at the word level.

**Baselines.** For our baselines, we compare against using the base (general-purpose, instruction-tuned) model to generate queries and also prompting the base model using bootstrapped few shot prompts optimized by DSPy. For the main few shot prompting baseline, we use the few shot prompts used during prompt driven diverse sampling. These prompts are created by optimizing the first hop with DSPy and then using that prompt for all hops. To demonstrate that the gains of LeReT are additive on top of these, we report the maximum achieved few-shot prompt performance achieved by *any* bootstrapped fewshot prompt $p_1, \ldots, p_P$. For some experiments, we also report few-shot all, which is where we use DSPy to optimize over the entire pipeline and bootstrap different examples for each hop. We also run Query2Doc (Wang et al., 2023), a prompting technique that asks the LLM to generate a hypothetical document in addition to the query, as a baseline. We also report LeReT-CD as a baseline for some experiments. This is the performance of the model after the SFT step (but before IPO) and as explained in Section 4.2 is the same as context distillation.

**Experiment setup.** Unless otherwise specified, we use Llama 3 8b Instruct or Gemma 2 9b it as the base model for query generation. We use ColBERTv2 (Santhanam et al., 2022) as the retriever. For the reward function, we use the average precision of the retrievals, so $\mathbb{R} = AP(C_{hi})$.

| Dataset | Model | Method | 1 Hop | | 2 Hops | | 3 Hops | | 4 Hops | |
|---------|-------|--------|-------|-------|--------|-------|--------|-------|--------|-------|
| | | | RE | AP | RE | AP | RE | AP | RE | AP |
| HotpotQA | Gemma 9b | Standard | 56.1 | 52.2 | 79.9 | 67.0 | — | | | |
| | | Iteration 1 | 55.7 | 51.7 | 78.2 | 65.6 | — | | | |
| | | Iteration 2 | 57.6 | 53.5 | **82.3** | 70.5 | — | | | |
| HoVer | Llama 8b | Standard | 45.8 | 42.5 | 65.4 | 56.1 | 72.8 | 61.4 | 76.9 | 64.3 |
| | | Iteration 1 | 45.1 | 41.7 | 62.7 | 54.1 | 69.6 | 59.1 | 73.5 | 62.1 |
| | | Iteration 2 | 44.9 | 41.6 | 65.3 | 55.5 | 73.4 | 61.2 | **78.2** | 64.4 |

Table 2: **Iteratively applying LeReT leads to performance gains compared to standard LeReT.** Gemma 9b and Llama 8b are tested with two iterations and recall and average precision are measured (higher is better).

## 5.1 RESULTS ON HOTPOTQA & HOVER

In terms of retrieval recall, LeReT improves recall by 9–22% on HotPotQA and 27–29% on HoVer relative to the Llama and Gemma unadapted instruct models ("base"). This substantially exceeds the gains achieved via few-shot prompting alone, showing that sampling from multiple few shot prompt ensembles and training the model with RL is crucial. The gains also compounds over hops, possibly because lower quality search queries at a given hop distract future steps. We feed the improved retrievals into Llama 3.1 70b, asking it to generate a response to the question using the provided context. We find that improving retrieval produces a corresponding improvement in generations, with the generator exact match increasing at approximately half the rate of recall.

## 5.2 ITERATIVE-LERET

We evaluate the performance of applying LeReT for two iterations. Training with only half the data (iteration 1) results in slightly worse performance compared to standard non-iterative LeReT, but after the second iteration the model performs better than the LeReT model. That is, sampling data that is both more on-policy and higher scoring in the second iteration leads to improvement.

## 5.3 FACTUALITY WITH DIFFERENT GENERATORS

| Generator Model | Base (RE 54.15) | | Few-shot (RE 63.60) | | LeReT (RE 80.40) | | |
|-----------------|------|------|------|------|------|--------|------|
| | EM | F1 | EM | F1 | EM | | F1 |
| Gemma 2b | 13.8 | 21.6 | 16.6 | 24.5 | 18.9 | (+5.1) | 28.5 |
| Llama 3 8b | 35.1 | 44.6 | 40.4 | 50.2 | 46.9 | (+11.8) | 58.6 |
| Llama 3.1 70b | 38.1 | 47.7 | 45.6 | 56.3 | 53.5 | (+15.4) | 64.9 |
| GPT4 | 33.4 | 43.3 | 41.6 | 52.7 | 50.7 | **(+17.3)** | 62.9 |

Table 3: **The stronger the generator model, the more it benefits from improved retrieval.** We test 4 different generator models using retrievals sampled from HotpotQA on the base model, few-shot prompted model, and LeReT-trained model. We report the recall of the retrievals (RE). We measure the exact match and F1 scores of the generated answers (higher is better).

Improving retrieval seeks to improve LLM grounding. Intuitively, stronger models with better reasoning capabilities should benefit more from having the correct documents to reason with than weaker models that may not be able to generate the correct answer even with the right documents. To assess this, we take the retrieval output by various Llama 3 8b models for HotpotQA and condition various generator models with them. As seen in Table 3, stronger generators deliver higher quality and larger gains when supplied with LeReT-trained retrieval contexts. We note that although GPT4 has the largest improvement, it does not have the highest score. Examining the generations, we see that GPT4 very closely followed the instructions to "base your answers only on the provided context" and would output statements such as "answer cannot be found in the context" instead of trying to answer it anyway the way weaker models did.

| Model | Data size | | Gold (%) | | Unique AP | | AP Std Dev | |
|---|---|---|---|---|---|---|---|---|
| | Hotpot | HoVer | Hotpot | HoVer | Hotpot | HoVer | Hotpot | HoVer |
| @ temp 0.3 | 44,705 | 28,575 | 37.5 | 12.1 | 1.66 | 1.25 | 0.09 | 0.029 |
| @ temp 0.5 | 55,564 | 32,091 | 39.4 | 12.6 | 2.10 | 1.27 | 0.10 | 0.033 |
| @ temp 0.7 | 69,361 | 37,354 | 43.9 | 12.6 | 2.10 | 1.28 | 0.14 | 0.033 |
| @ temp 1.35 | 92,264 | 48,516 | 46.3 | 12.0 | 2.35 | 1.32 | 0.16 | 0.039 |
| @ temp 2.0 | 81,017 | 58,281 | 41.0 | 10.3 | 2.36 | 1.34 | 0.16 | **0.045** |
| Fixed few-shot @ temp 2.0 | 93,613 | **92,542** | 47.7 | 13.5 | 2.41 | 1.32 | 0.16 | 0.038 |
| Diverse few-shot @ temp 2.0 | 95,373 | 71,433 | 47.5 | 12.3 | **2.47** | **1.34** | **0.17** | 0.044 |
| Diverse few-shot @ temp 0.7 | **105,506** | 49,304 | **54.2** | **13.8** | 2.35 | 1.28 | 0.15 | 0.031 |

Table 4: **Sampling with higher temperature results in greater diversity of responses (higher unique ap, ap std dev) while few shot prompting results in better data (higher gold rate).** Specifically, gold rate is defined as the percentage of questions for which we have a gold star response (a query that results in the maximal score). Unique AP is the number of unique average precision scores there are for a question, and AP std dev is the standard deviation of the average precision scores for a question. Data size is measured in terms of the number of preference pairs.

## 5.4 DIVERSE FEW-SHOT PROMPTING VERSUS HIGH-TEMPERATURE SAMPLING

While prior work typically uses high temperature sampling to generate diversity, LeReT leverages few-shot prompting to generate diverse exploration data. To evaluate the effectiveness of few-shot prompt diversification, we consider alternate sampling strategies, particularly sampling from $\pi_r$ at different temperatures with no few-shot prompting, a fixed few-shot prompt (fixed), or multiple few shot prompts (diverse).

First, we compute some statistics about the rewards generated by the sampled search queries: we compute the average number of unique rewards per question and the standard deviation of the rewards as a proxy for diversity, and we measure the percentage of questions where at least one query (gold star answer) achieves maximal reward as a proxy for quality of sampled data. We find in Table 4 that while sampling with higher temperature improves diversity in search queries, few-shot prompting leads to significantly higher quality data and using multiple few-shot prompts provides comparable diversity.

We train on four of the sampled datasets: (1) queries sampled with diverse few-shot prompting at standard temperature (0.7), (2) queries sampled at a high temperature (2.0), (3) queries sampled with diverse few-shot prompting at high temperature (2.0), and (4) queries sampled with a single fixed few-shot prompt at high temperature (2.0). We find that training a model using data sampled with few-shot prompting at temperature 0.7 results in the best performance, which is also the sampling strategy that results in the largest percentage of questions with at least one gold star answer. This suggests that exploration is critical to the success of RL training and justifies the extra effort of bootstrapping few shot prompts.

| Model | 1 Hop | | 2 Hops | |
|---|---|---|---|---|
| | RE | AP | RE | AP |
| Base model | 42.3 | 38.8 | 54.7 | 41.9 |
| Few-shot | 49.87 | 45.58 | 64.77 | 53.86 |
| LeReT diverse few-shot @ temp 0.7 | **55.74** | 51.76 | **78.14** | 67.67 |
| LeReT @ temp 2.0 | 49.12 | 44.81 | 69.86 | 58.56 |
| LeReT fixed few-shot @ temp 2.0 | 50.89 | 46.69 | 70.79 | 59.93 |
| LeReT diverse few-shot @ temp 2.0 | 51.73 | 47.79 | 73.67 | 62.60 |

Table 5: **Diversifying few-shot prompts when sampling search queries result in more effective training datasets for RL.** The standard LeReT with few-shot prompting at temperature 0.7 results in better performance than training on a dataset sampled at temperature 2.0 without few-shot prompting or a dataset sampled at at temperature of 2.0 with fixed few-shot prompting or a dataset sampled at at temperature of 2.0 with diverse few-shot prompting.

| Dataset | Method | 1 Hop | | 2 Hops | | 3 Hops | | 4 Hops | |
|---------|--------|-------|------|--------|------|--------|------|--------|------|
| | | RE | AP | RE | AP | RE | AP | RE | AP |
| Hotpot | Llama 8b | 10.6 | 9.1 | 18.5 | 15.0 | | — | | |
| | Few-shot | 39.6 | 34.7 | 50.7 | 40.8 | | — | | |
| | LeReT | 43.8 | 38.9 | **60.0** | 48.6 | | — | | |
| HoVer | Llama 8b | 15.6 | 14.0 | 22.8 | 19.2 | 27.8 | 22.0 | 31.2 | 23.5 |
| | Few-shot | 37.1 | 33.2 | 45.9 | 38.3 | 49.8 | 40.1 | 52.1 | 39.6 |
| | LeReT | 39.1 | 35.2 | 51.9 | 44.5 | 58.6 | 48.8 | **62.6** | 51.3 |

Table 6: **LeReT greatly improves the performance of Llama 8b on Hotpot and HoVer with Azure AI Search used as the retriever.** We perform the same sampling and training pipeline as all other experiments but use Azure AI Search instead of ColBERT.

## 5.5 DIFFERENT RETRIEVERS

Finally, we test whether LeReT is applicable to general RAG systems by swapping our retriever from ColBERT over Wikipedia to Azure AI Search, applied with the default configuration for full text search. We observe that the base Llama model performs very poorly compared to its retrievals with ColBERT. This is likely because Azure is not specialized to the Wikipedia index which is helpful for our multi-hop tasks. However, the query generating LLM can adapt to compensate for this weaker retriever, as we see significant improvement with few-shot prompting and LeReT. This demonstrates the power of LeReT to adapt to general blackbox tools in the pipeline. Given the poor performance of the base model, few-shot prompting based exploration (Section 5.4) is found to be necessary, versus simply sampling with high temperature.

## 6 DISCUSSION

In this work, we introduced LeReT, a framework for improving the quality of multi-hop retrieval via reinforcement learning and thus enabling better grounding for LLM systems. Beyond retrieval specifically, this can be extended to learning for agentic systems or LLM programs that use other tools. LeReT conducts diversified exploration of search queries in each hop by sampling using varied optimized few-shot prompts. It then uses this to construct a preference dataset for every hop consisting of queries that lead to a diverse set of retrieved outcomes. To train the model, it first conducts context distillation followed by an iterative application of the IPO objective. Experimental evaluation on the HotpotQA and HoVer benchmarks with two different retrieval models reveals that LeReT can improve the quality of Llama 8b- and Gemma 9b-based systems by up to 29% in recall.

**Limitations & Future Work.** While in this work we have used direct supervision for retrieval, a fruitful effort would be to enable learning from indirect supervision such as the correctness of the final generative response. Another promising direction is learning by updating the tools themselves like training the retriever model used to encode the search queries and documents. Doing this would require changes to the sampling algorithm and addressing the signal-to-noise ratio but would likely lead to significant gains.

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

## A    SAMPLING AND TRAINING DETAILS

We sample with $P = 4$ for HotpotQA and $P = 3$ for HoVer. We implement our sampling pipeline on top of DSPy (Khattab et al., 2023), specifically defining a single hop as a program and sampling data using the evaluate functions. We also use chain-of-thought prompting when generating queries.

We use a learning rate of `1e-7` for SFT/context distillation in all our experiments, and use a $\tau = 0.05$ and learning rate of `1e-7`. We train SFT for 1 epoch, and we only distill the best performing prompt. We train IPO for 2 epochs.

### A.1    DATA SCALING ANALYSIS

We conduct data scaling experiments for LeReT. We evaluated a training run of Llama 3 8b on the full HotpotQA training set (90,447 questions), which resulted in 494,208 preference pairs after prompt driven diverse sampling. We find that the majority of the improvement occurs relatively quickly. Based on this, we only use a quarter of the HotpotQA training set for subsequent experiments. However, data scaling likely depends on a host of factors, including the task complexity and the base model, and we conducted the data scaling experiment to reduce the computational cost of our experiments.

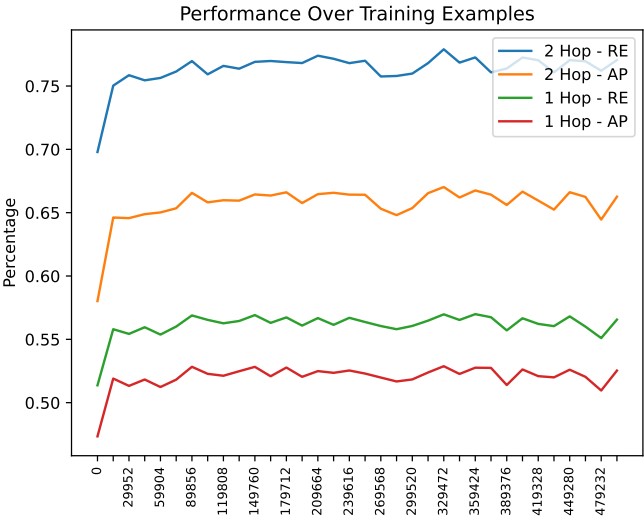

Figure 4: **The model performance saturates quickly**. Measuring the test performance of Llama 3 8b as training progresses on the preference dataset collected using LeReT on the full HotpotQA train set (90,447 HotpotQA questions, 494,208 preference pairs).

For the retriever, we set up a local instance of ColBERTv2 with the Wiki 2017 abstracts index. For Azure AI Search, we created a custom index by uploading all the abstracts from the Wiki 2017 abstracts index. We use the default settings, only using standard text search with no semantic or vector search.

## B    ADDITIONAL EXPERIMENTAL DETAILS

### B.1    JUSTIFYING GREEDY OPTIMIZATION

We run LeReT for two hops on all 90,447 questions in HotpotQA. After sampling the first hop, we chose two sets of documents that have different rewards. We then run the second hop with no few-shot prompting and evaluate the reward after the second hop. We find that the lower reward set of documents resulted in a higher reward after the second hop in only 0.026% of cases.

## B.2 Average precision

Average precision is defined according to Eq. 3 where $R$ is the total number of relevant documents, $P(k)$ is the precision of the first $k$ documents, and rel$(k)$ is 1 if the $k$th document is relevant and 0 otherwise:

$$\text{AP} = \frac{1}{R} \sum_{k=1}^{N} P(k) \cdot \text{rel}(k) \tag{3}$$

## B.3 Multi-Hop Sampling

Can we get away without training on data from all hops? We run an ablation to determine the necessity of sampling across multiple hops. Sampling across multiple hops requires is computationally and less parallelizable. Specifically, we train Llama 3 8b on Hotpot and HoVer, sampling only the data from the first half of the hops. For Hotpot, this amounts to sampling from just the first hop and for HoVer, this amounts to sampling data from the first two hops.

| Dataset | Method | 1 Hop | | 2 Hops | | 3 Hops | | 4 Hops | |
|---|---|---|---|---|---|---|---|---|---|
| | | RE | AP | RE | AP | RE | AP | RE | AP |
| Hotpot | 1 hop | 58.8 | 54.7 | 70.8 | 62.5 | — | | | |
| | All hops (2) | 56.7 | 52.5 | **77.1** | 66.3 | — | | | |
| HoVer | 2 hops | 45.5 | 42.2 | 62.8 | 54.3 | 69.7 | 59.3 | 73.6 | 62.2 |
| | All hops (4) | 45.8 | 42.5 | 65.4 | 56.1 | 72.8 | 61.4 | **76.9** | 64.3 |

Table 7: **Sampling from all hops instead of only the first few significantly improves performance.** We train Llama 3 8b on preferences from only the first half of the hops in a pipeline, and compare to training on the full dataset containing preferences over queries from all the hops.

We find training on data from all hops leads to substantial improvement in performance across both the datasets, compared to training on only the first half leads to performance gains. The gain is larger for Hotpot where the task only has two hops, and thus never sees data where the context has additional documents from previous hops.

## B.4 Long form generations

We present a preliminary attempt at long form generation. The majority of LLM use cases are for long form generation, and as such we want to test LeReT's ability to improve long form generations. In addition, it is more to difficult to evaluate the factuality of long form answers, meaning that evaluating the relevance of the documents it conditioned its answer on as LeReT does may be simpler. Current retrieval datasets focus on short question answering, leading us to generate our own long form dataset.

To create a dataset with open ended questions that still had correct retrievals, we prompted GPT for 20 broad topics. From each of those 20 broad topics, we prompted GPT for 500 topics, giving us a total of 10,000 topics such as "Injuries in American football" or "Effects of mobile radiation on human health". We then fed those topics into Colbert and retrieved the top 10 wikipedia abstracts. We then prompt GPT with the 10 wikiepdia abstracts and asked it to come up with a question that required students to use exactly 3 of the 10 articles. This gave it the freedom to choose articles that were closely related and led to more natural questions than forcing it to use a given 3 articles.

We then train Llama 3 8b on this dataset using LeReT. We find an approximately 8.47% improvement in document retrieval. We create long form generations by feeding the retrieved documents into Llama 3.1 70B. We find that the LeReT-generations are superior to few shot prompting with a 55.56% win rate.

## B.5 Possible reward functions

| Dataset | Disagree (%) ↓ | | Data Size ↑ | |
|---|---|---|---|---|
| | **Hard** | **Soft** | **Generator** | **Retrieval** |
| Hotpot | 25.09 | 38.46 | 96,766 | 163,644 |
| Hover | 31.48 | 33.03 | 10,488 | 88,448 |

Table 8: **Sampling data using the generator F1 score leads to poor quality data with many wrong preference pairs (high disagree values) and less data (lower data size).** We sample preference datasets using the F1 score of the generated answer. Hard disagree is preference pairs where the ranking by the retrieval reward is swapped compared to the ranking by the generator reward. Soft disagree is preference pairs where the retrieval reward is equal but the generator reward has a ranking. Data size is the size of the preference dataset generated using each method.

Do we need direct supervision in LeReT for computing the reward function that outputs a reward for a given question and set of retrieved documents or can we get away with indirect methods for supervising retrieval quality? We currently use average precision of retrieved documents, which provides more direct supervision for retrieval but requires knowing a correct set of documents in advance, as is available in Hotpot/HoVer. But, there are settings where the optimal retrieved documents may be hard to specify in advance. In such cases, indirect supervision may be easier to provide, where the final generated answer conditioned on the retrieved documents is reviewed, and a reward is generated based on the verification of the final answer. Such supervision can be quite weak and have a high amount of noise, as the generator may answer correctly even when conditioned on incorrect documents (because of internal knowledge) or provide incorrect answers even when conditioned on the right documents.

To explore these different settings, we apply LeReT using the F1 score of the final generated answer as the reward. We condition the LLM on retrieved documents along with the actual question to generate an answer, and compare that to the correct answer. Formally, instead of using $\mathrm{AP}(C_{hi})$ as the reward, we use $\mathbb{R} = F1(\pi_g(d, C_{hi}))$.

We find that the F1 score of the generator does not provide a very strong signal. For the preference dataset that we constructed using the generator, over 50% of the preference pairs are wrong. We split these incorrect pairs into two categories: hard and soft disagree. Hard disagree means that according to the F1 generation reward, $q_i \succ q_j$ but according to the AP retrieval reward, $q_i \prec q_j$. Soft disagree means that according to the F1 generation reward, $q_i \succ q_j$ but according to the AP retrieval reward, $q_i = q_j$. To reduce the number of soft disagrees, we experiment with adding a threshold for the difference in F1 score to form a preference pair, but find that over half the questions in HotpotQA and all the questions in HoVer have one word answers so this is not effective.

| Model | 1 Hop | | 2 Hops | |
|---|---|---|---|---|
| | **RE** | **AP** | **RE** | **AP** |
| Base model | 44.2 | 40.6 | 55.5 | 43.0 |
| LeReT-Retriever | 56.7 | 52.5 | 77.1 | 66.3 |
| LeReT-Generator | 49.6 | 45.5 | 64.5 | 53.8 |

Table 9: **With the weaker signal of generation, LeReT is able to improve upon the base model but does not match the performance of using the retriever reward.** We take the dataset sampled on Hotpot and train Llama 3 8b.

We also find that the datasets generated using the generator reward are significantly smaller than those generated using the AP retriever reward, for example about 8.4 times smaller in case of HoVer. Since HoVer has one word answers, the generator F1 score is less fine grained than the retrieval accuracy over 4 documents. This leads to a smaller preference dataset as LeReT provides contexts to the next hop only if they have not achieved the maximum score. In the one-word answer case, there could be queries that retrieve only some (or none) of the correct documents but give the generator enough context to guess or use prior knowledge to output the correct answer. Thus, with this sort of coarse reward, it is much more likely that a question will be excluded from subsequent hop even though the model has not output an optimal query to the retriever.

When we train a model on this data, we find that it significantly improves over the base model, but substantially under performs the case where the reward signal is derived from average precision of gold documents. We find that the first hop data is far better than the second hop data, (11.43% hard disagree compared to 37.70, 37.04 soft disagree compared to 39.78) which likely contributes to the decreased performance on the second hop.

## B.6    GENERALIZATION

We test how well LeReT-trained models are able to generalize. Specifically, we take Gemma 9b trained on Hover and evaluate its performance on Hotpotqa. Similarly, we take Llama 8b trained on Hotpotqa and test in on Hover. We find that while these models do not perform as well as those specifically trained on the dataset, they outperform few-shot prompting and as such with LeReT, models are learning to search in ways that are broadly applicable to different datasets.

| Test Dataset | Model | Method | 1 Hop | | 2 Hops | | 3 Hops | | 4 Hops | |
|---|---|---|---|---|---|---|---|---|---|---|
| | | | RE | AP | RE | AP | RE | AP | RE | AP |
| HotpotQA | Gemma 9b | Few-shot | 49.9 | 45.6 | 64.8 | 53.9 | — | | | |
| | | LeReT-Hotpot | 56.7 | 52.5 | 77.1 | 66.3 | — | | | |
| | | LeReT-Hover | 51.3 | 47.2 | 73.8 | 60.7 | — | | | |
| HoVer | Llama 8b | Few-shot | 45.6 | 42.2 | 53.4 | 45.9 | 56.0 | 46.0 | 57.3 | 46.1 |
| | | LeReT-Hover | 45.8 | 42.5 | 65.4 | 56.1 | 72.8 | 61.4 | 76.9 | 64.3 |
| | | LeReT-Hotpot | 42.6 | 39.3 | 60.1 | 51.6 | 67.5 | 56.6 | 71.2 | 59.2 |

Table 10: **Models trained with LeReT on a given dataset lead to improved performance on the other dataset.** Gemma 9b trained on Hover is tested on Hotpot and Llama 8b trained on Hotpot is tested on Hover. Both models outperform few shot prompting.

