# OpenReview forum: "Grounding by Trying: LLMs with Reinforcement Learning-Enhanced Retrieval"
_ICLR.cc/2025/Conference — ICLR 2025 Poster_

### Official Review · Reviewer_9WtD · 2024-10-22

**Soundness:** 3
**Presentation:** 2
**Contribution:** 3
**Rating:** 6
**Confidence:** 4

**Summary:**

The paper addresses the issue of hallucination in language models by enhancing their capability to generate effective queries for retrieving relevant facts, thereby improving the accuracy of model responses. The proposed approach, LeReT, generates diverse search queries by incorporating few-shot examples. It constructs comparative queries guided by the supervision of retrieval quality and optimizes the model using the IPO algorithm. The study empirically validates the LeReT framework, exploring various methods of collecting retrieval rewards and assessing performance across different retrievers.

**Strengths:**

1. The motivation that enhancing RAG performance through learning to generate better search queries is clear and reasonable.
2. The experiments present the improvement in both retrieval quality and finally performance, which validate the soundness of proposed algorithm.

**Weaknesses:**

1. The writing and presentation of results in this work could be enhanced. On line 254, I think it should be, "In this work, we consider XXX."

2. Some baselines (https://aclanthology.org/2023.emnlp-main.585.pdf, https://openreview.net/pdf?id=vDvFT7IX4O, and https://aclanthology.org/2023.acl-long.99.pdf) could be added to both the experiments and the related works sections. Although these works may not directly focus on multi-hop retrieval, they fit broadly within the same topic as this work, i.e., "query expansion." These works focus on generating better search queries to enhance RAG performance.

3. Relying on annotated golden documents, referred to as "direct supervision" in this work, limits the vision of the study. In more general scenarios, collecting "indirect supervision" is more feasible. Moreover, signals from "indirect supervision" are the ultimate indicators for downstream tasks.

**Questions:**

1. On lines 218–219, are there any concrete examples illustrating the differences between BFRS-generated queries and high-temperature sampling queries?

2. Regarding line 260, what are the examples used in few-shot prompting? Why is few-shot prompting not feasible during testing? Is it due to considerations of inference efficiency?

3. As for section 4.3, are there any details about how rewards are computed?

4. For Iterative-LeReT and the discussion in Section 4.3, this multi-hop search process seems compatible with online RL training, such as using PPO, by incorporating process rewards at each hop and a final reward for the conclusion. Have the authors explored online RL training?

---

> ### Author Response · Authors · 2024-11-22
>
> We thank reviewer 9WtD for their detailed feedback and suggestions, especially around baselines!
>
> We have revised the paper and improved the writing.
>
> > Some baselines (https://aclanthology.org/2023.emnlp-main.585.pdf, https://openreview.net/pdf?id=vDvFT7IX4O, and https://aclanthology.org/2023.acl-long.99.pdf) could be added to both the experiments and the related works sections. Although these works may not directly focus on multi-hop retrieval, they fit broadly within the same topic as this work, i.e., "query expansion." These works focus on generating better search queries to enhance RAG performance.
>
> We appreciate reviewer 9WtD bringing our attention to these related works, and we have updated our related works section appropriately. We have added Query2Doc[1] as a baseline in Table 1. To summarize the discussion here: Tree of Clarifications[2] is a promising method, as it performs question clarification after retrieval in order to improve answer generation. Since LeReT mainly focuses on improving retrieval, the work is complementary to our work and Tree of Clarifications could be run on LeReT-improved retrievals. While HyDE seems quite effective, we find that it requires an additional contrastive encoder to find similar documents to the generated hypothetical document. Given that LeReT is targeting the common use case of a black box retriever like the Bing API or Azure AI Search where it is not possible to access an encoder as required by HyDE, it is not possible to compare the two methods. Additionally, the Query2doc baseline already measures the performance gains from hypothetical documents in the black box retriever setting. We would also like to emphasize that LeReT, our reinforcement-learning based framework for improving retrieval, can be combined with these prior works. For example, LeReT could be used to sample hypothetical documents and queries generated by Query2doc or HyDE and fine tune the model to produce better documents and queries. We are excited for future work to explore these directions.
>
> 1. Query2doc: Query Expansion with Large Language Models. Liang Wang, Nan Yang, Furu Wei. Empirical Methods in Natural Language Processing (EMNLP) 2023.
> 2. Tree of Clarifications: Answering Ambiguous Questions with Retrieval-Augmented Large Language Models. Gangwoo Kim, Sungdong Kim, Byeongguk Jeon, Joonsuk Park, Jaewoo Kang.  Empirical Methods in Natural Language Processing (EMNLP) 2023.
> 3. Precise Zero-Shot Dense Retrieval without Relevance Labels. Luyu Gao, Xueguang Ma, Jimmy Lin, Jamie Callan. Association for Computational Linguistics (ACL) Anthology, 2023.
>
> > Relying on annotated golden documents, referred to as "direct supervision" in this work, limits the vision of the study. In more general scenarios, collecting "indirect supervision" is more feasible. Moreover, signals from "indirect supervision" are the ultimate indicators for downstream tasks.
>
> We agree that using LeReT with indirect supervision would allow the method to be used in more general scenarios. However, we find in the data scaling curves in Appendix A.1 that relatively few data points are needed to achieve a lot of the performance gains, so even with limited data LeReT is likely to be effective. In addition, the trade-off in feasibility between collecting “indirect supervision” and “direct supervision”  is often not clear. There are scenarios where collecting direct supervision by judging between two sets of documents might be easier than collecting indirect supervision by comparing the generated answers. For example, annotators could judge whether a LLM is sourcing medical advice from reliable sites without necessarily having the medical knowledge to evaluate the generated medical information.

---

> ### Author Response · Authors · 2024-11-22
>
> > On lines 218–219, are there any concrete examples illustrating the differences between BFRS-generated queries and high-temperature sampling queries?
>
> It is hard to empirically observe any differences between queries, especially in single examples. However, overall few shot prompting seems to lead to more consistent queries while high-temperature sampling will sometimes lead to differently formatted outputs like SQL queries instead of a phrase you might type into a retriever.
>
> > Regarding line 260, what are the examples used in few-shot prompting? Why is few-shot prompting not feasible during testing? Is it due to considerations of inference efficiency?
>
> The examples used in few-shot prompting are question and query pairs that result in good retrievals selected using DSPy’s bootstrapping. Few-shot prompting is feasible during testing, however part of our recipe is to perform context distillation before preference optimization via IPO. Additionally, we use different sets of few shot prompts and want to teach the model to output the best query generated by any of the few shot prompts. To do this during test time would involve ensembling across all the few shot prompts, and as demonstrated by our empirical results in Table 1, preference optimization leads to much larger gains than ensembling across few-shot prompts. It would be interesting for future work to combine few-shot prompting with preference-optimized models from LeReT.
>
> > As for section 4.3, are there any details about how rewards are computed?
>
> Yes, for direct supervision we use the average precision of the retrieved documents versus correct documents. For indirect supervision, we experimented with the F1 score of the generated answer as detailed in Appendix B.5 and recently also experimented with providing the generator with the correct answer and asking it which set of retrieved documents would be more helpful.
>
> > For Iterative-LeReT and the discussion in Section 4.3, this multi-hop search process seems compatible with online RL training, such as using PPO, by incorporating process rewards at each hop and a final reward for the conclusion. Have the authors explored online RL training?
>
> Iterative-LeReT can be seen as a form of online RL training. LeReT is compatible with any RL algorithm and we agree that further exploration into PPO or other algorithms could be fruitful.

---

> > ### Comment · Reviewer_9WtD · 2024-11-25
> >
> > The authors have largely addressed my concerns, and I acknowledge the reliability of the motivation and methodology of this paper. Therefore, I have decided to raise my score.
> > However, I would like to add that I do not think it is appropriate to refer to Iterative-LeReT as online RL training. Additionally, training PPO using both process rewards at each hop and a final reward for the conclusion seems to me to be a more direct approach.

---

### Official Review · Reviewer_cwbp · 2024-11-01

**Soundness:** 2
**Presentation:** 3
**Contribution:** 2
**Rating:** 6
**Confidence:** 3

**Summary:**

The paper presents LeReT, a framework for improving LLM retrieval and answer grounding using reinforcement learning. It uses prompt-driven diverse query generation, model optimization with preference-based RL, and reward labeling for retrieved documents. Experiments on HotpotQA and HoVer datasets show significant improvements in retrieval and downstream generation compared to baselines. Analysis reveals the importance of diverse few-shot prompting and LeReT's applicability across different retrievers. Limitations include the use of direct supervision, and future work may focus on indirect supervision and tool updating.

**Strengths:**

- LeReT is applicable to general retrieval-augmented generation (RAG) systems and can adapt to different retrievers.
- LeReT significantly improves retrieval accuracy. Compared to the unadopted Llama and Gemma instruction models, the recall rate increases by 9-22% on HotPotQA and 27-29% on HoVer.
- It can be used iteratively: applying LeReT for two iterations shows that the model performance after the second iteration is better than that of the standard non-iterative LeReT.

**Weaknesses:**

- The novelty assertion of the proposed method lacks clarity. Regarding the related work spanning from line 139 to 148, it remains ambiguous as to how the proposed method differentiates itself from those other methods. I comprehend that the proposed approach employs diverse query generation and IPO for preference learning. However, these seem to be more of incremental enhancements within an existing framework rather than representing a distinct novelty.
- Also, the experiments lack comparisons with the most relevant recent works. Only the basic few-shot prompt baseline is compared.

**Questions:**

- Line 76 "If LLMs can observe the retrieved documents for different search queries, they can learn which queries lead to better outcomes." What supports this claim? And if this is true, then why do you need the direct or indirect supervision to teach LLM "how to query"?

---

> ### Author Response · Authors · 2024-11-22
>
> We thank reviewer cwbp for their thoughtful comments and suggestions!
>
> > The novelty assertion of the proposed method lacks clarity. Regarding the related work spanning from line 139 to 148, it remains ambiguous as to how the proposed method differentiates itself from those other methods. I comprehend that the proposed approach employs diverse query generation and IPO for preference learning. However, these seem to be more of incremental enhancements within an existing framework rather than representing a distinct novelty.
>
> Thank you for raising this point and we have further updated our related work to clarify our contributions.  Recent work has focused on creating better retrievers (RAG[1], DPR[2], ColBERT[3], Baleen[4]), prompt-based query generation techniques (Query2Doc[5], Rethinking with Retrieval[6], DSPy[7]), and downstream answer generation methods (Chain of Note[8], Copy is All You Need[9], Tree of Clarifications[10]). In comparison, our work demonstrates how RL can be used to improve modern retrieval systems effectively. We propose prompt-driven diverse query generation, which enables RL to effectively improve LLMs’ retrieval capabilities. Our empirical results show that RL substantially improves retrieval and downstream performance beyond baselines (see Table 1).
>
> Beyond any specific framework, LeReT is broadly generalizable and can be applied on top of many various existing methods. LeReT can be used with prompting-based approaches such as hypothetical document generation by collecting data on which hypothetical documents / more broadly the output of the prompt lead to better results and teaching the model to do that. In addition, as LeReT treats the retriever as a black box, it can be used with ongoing improvements to actual retrieval systems.
>
> 1. Retrieval-Augmented Generation for Knowledge-Intensive NLP Tasks. Patrick Lewis et al. NeurIPS 2020.
> 2. Dense Passage Retrieval for Open-Domain Question Answering. Vladimir Karpukhin et al. EMNLP 2020.
> 3. ColBERT: Efficient and Effective Passage Search via Contextualized Late Interaction over BERT. Omar Khattab et al. SIGIR 2020.
> 4. Baleen: Robust Multi-Hop QA Dataset via Retrieval Augmentation. Xiang Lisa Li et al. NeurIPS 2021.
> 5. Query2Doc: Query Expansion with Large Pre-trained Language Models. Jiaxin Mao et al. SIGIR 2021.
> 6. Rethinking with Retrieval: Faithful Large Language Model Inference. Hangfeng He et al. ACL 2023.
> 7. DSPy: Compiling Declarative Language Model Calls into State-of-the-Art Pipelines. Omar Khattab et al. ICLR 2024.
> 8. Chain-of-Note: Enhancing Robustness in Retrieval-Augmented Language Models. Wenhao Yu et al. EMNLP 2024.
> 9. Copy Is All You Need. Tian Lan et al. ICLR 2023.
> 10. Tree of Clarifications: Answering Ambiguous Questions with Retrieval-Augmented Large Language Models. Gangwoo Kim et al. EMNLP 2023.
>
> > Also, the experiments lack comparisons with the most relevant recent works. Only the basic few-shot prompt baseline is compared.
>
> Thanks for raising this concern. We have added comparisons to Query2Doc [1], and have added a discussion around other related methods for retrieval. We believe our contribution is compatible / orthogonal to several other works, thus we compare to key prior algorithmic works. What other baselines do you think are meaningful to compare LeReT to?
>
> [1] Query2doc: Query Expansion with Large Language Models. Liang Wang, Nan Yang, Furu Wei. Empirical Methods in Natural Language Processing (EMNLP) 2023.
>
> > Line 76 "If LLMs can observe the retrieved documents for different search queries, they can learn which queries lead to better outcomes." What supports this claim? And if this is true, then why do you need the direct or indirect supervision to teach LLM "how to query"?
>
> We apologize for the ambiguity and have revised the paper accordingly (line 45). We meant that by observing the retrieved documents for different search queries, we can generate a reward for the query, and train the model to output queries that get a high-reward.

---

> > ### Comment · Reviewer_cwbp · 2024-11-26
> >
> > This response has mostly address my concerns so I gave a score raise.

---

### Official Review · Reviewer_v1Yv · 2024-11-04

**Soundness:** 3
**Presentation:** 3
**Contribution:** 3
**Rating:** 6
**Confidence:** 3

**Summary:**

This paper presents a framework called "Learning to Retrieve by Trying" (LeReT), which aims to improve the grounding of large language models (LLMs) through reinforcement learning (RL)-based retrieval. LeReT enables LLMs to generate queries, learn from trial-and-error, and enhance retrieval by using diverse few-shot prompts combined with preference-based reinforcement learning. LeReT's flexibility makes it adaptable across different retrieval systems, with potential applicability in broader retrieval-augmented generation (RAG) contexts.

**Strengths:**

1. Introduces a unique reinforcement learning framework to improve retrieval accuracy in LLMs, especially for complex multi-hop queries.
2. Demonstrates the effectiveness of iterative training in enhancing the retrieval and grounding abilities of LLMs.
3. Compatible with various retrieval systems, including ColBERTv2 and Azure AI Search, indicating its broad applicability.

**Weaknesses:**

1. Primarily relies on direct supervision for labeling relevant documents, which may limit its scalability in cases where explicit relevance labels are unavailable.
2. Requires extensive computation due to multi-hop retrieval and diverse query sampling, making it resource-intensive.
3. The need for sampling across multiple hops is computationally intensive and less parallelizable, reducing scalability.

**Questions:**

1. While iterative training improves retrieval accuracy, does it potentially overfit the model to specific multi-hop tasks, thereby impacting generalizability in other retrieval-augmented scenarios?
2. What is involved in adapting LeReT to new domains or types of queries that it was not originally trained on?

---

> ### Author Response · Authors · 2024-11-22
>
> We thank reviewer v1Yv for their thoughtful comments and insights, particularly into the scalability and generalization of our work!
>
> > Primarily relies on direct supervision for labeling relevant documents, which may limit its scalability in cases where explicit relevance labels are unavailable.
>
> We agree that adapting LeReT for indirect supervision is an exciting direction and would improve the scalability of the method. A couple of salient points:
> - As the data scaling curves in Appendix A.1 suggest, a lot of the performance gains happen with relatively few data points, so LeReT should provide meaningful performance improvements even when limited data is available.
> - The trade-off between comparing final answers (i.e. indirect supervision) and retrieved documents (i.e. direct supervision) is often not clear. There are situations where judging between two sets of documents might be easier than comparing the generated answers. For example, annotators could judge whether a LLM is sourcing medical advice from reliable sites without necessarily having the medical knowledge to evaluate the generated medical information.
>
> > Requires extensive computation due to multi-hop retrieval and diverse query sampling, making it resource-intensive.
> > The need for sampling across multiple hops is computationally intensive and less parallelizable, reducing scalability.
>
> We agree that LeReT is resource-intensive. However, as demonstrated by the scaling curve in Appendix A.1, the majority of the improvement is realized with relatively few preference pairs. Moreover, the trained model can be deployed cheaply compared to other methods like few-shot prompting that require additional tokens during inference, Query2doc which requires multiple calls to the LLM, and Tree-of-Clarification which makes many recursive calls to the generator. One-time training cost may be very favorable in several use cases given the low cost at the time of inference.
>
> > While iterative training improves retrieval accuracy, does it potentially overfit the model to specific multi-hop tasks, thereby impacting generalizability in other retrieval-augmented scenarios?
> > What is involved in adapting LeReT to new domains or types of queries that it was not originally trained on?
>
> We agree that overfitting is a potential concern. Empirically, we see from looking at generated queries that general things such as learning not to output SQL queries but instead phrases can be generalized across domains. We also test a model trained on Hover on HotpotQA (and vice versa) and find that while these models do not perform as well as models trained on the dataset, they outperform few-shot prompting. Full experimental results are available in the now updated Appendix B.6. Algorithmically, IPO is more resistant to overfitting. We find that we can generalize our retrieval models from one domain to another. Building on modern LLMs enables such generalization to other domains as pre-training happens on extremely broad and diverse datasets.

---

> > ### Comment · Area_Chair_MKpG · 2024-11-26
> >
> > Dear Reviewer v1Yv the ICLR discussion period is extended. Could you please take a look at the authors' rebuttal and other reviews, and see whether you would like to update your ratings? The authors would greatly appreciate your consideration and responses.

---

### Official Review · Reviewer_NW9U · 2024-11-04

**Soundness:** 3
**Presentation:** 4
**Contribution:** 3
**Rating:** 6
**Confidence:** 4

**Summary:**

In multi-hop question answering, a model needs to perform multiple retrieval steps before arriving at an answer.  Since the reward (answer) is not known until the last step, this problem lends itself well to reinforcement learning (RL).  The paper proposes to optimize one component (the question generator for retrieval) in the multi-hop QA pipeline using RL.  The method uses direct supervision (gold retrieved docs) to first train a reward function.  Then a diverse set of queries are sampled from the model using varied prompts.  The rewards for these generations are fed into an RL algorithm (IPO) to improve the query generator.  The method improves substantially on pure SFT methods on HotpotQA and HoVer.

**Strengths:**

* Well written paper, clear presentation
* Results are strong and convincingly support the claim that using RL to improve query generator works better than pure SFT.

**Weaknesses:**

* A bit difficult to judge the novelty of the contribution.  Has similar methods been used for single-hop QA or RAG in general?  If so, the novelty here might be marginal, especially since the paper relies on direct supervision for the reward model.
* The two multi-hop datasets used are not natural and somewhat out-of-date.  It would be great to see the methods usefulness on more relevant tasks and benchmarks.  The long-form generation attempt in the appendix is interesting, and could perhaps be developed more and moved into the main text?

**Questions:**

* Has similar methods been used for single-hop QA or RAG in general?

---

> ### Author Response · Authors · 2024-11-22
>
> We thank reviewer NW9U for their thorough summary and feedback!
>
> > A bit difficult to judge the novelty of the contribution. Has similar methods been used for single-hop QA or RAG in general? If so, the novelty here might be marginal, especially since the paper relies on direct supervision for the reward model.
>
> Thanks for raising this point and we recognize that there are many contributions on this topic. We believe [1] is closest to our framework, making comparable assumptions related to direct supervision as our work does. We have updated the related work with a discussion around this. There are several important differences that are crucial to LeReT, especially given the advances in LLMs since 2017. First, [1] does not use a generative LM but a query selection architecture (understandably, since it is 2017). Importantly, a straightforward application of RL similar to [1] for modern generative LLMs would not perform better than  few-shot prompting with current LLMs, as seen in Table 5 when comparing few-shot and LeReT @ temp 2.0. Our proposed prompt-driven diverse query generation is what allows reinforcement learning to be effective and improve current LLMs for retrieval. At a fundamental level, our strategy leverages advances in modern LLMs to improve the exploration for the RL problem and leverage the generative nature of these LLMs, substantially improving the downstream performance beyond what few-shot prompting can achieve.
>
> [1] Task-Oriented Query Reformulation with Reinforcement Learning. Rodrigo Nogueira, Kyunghyun Cho. Association for Computational Linguistics (ACL) Anthology, 2017.
>
> > The two multi-hop datasets used are not natural and somewhat out-of-date. It would be great to see the methods usefulness on more relevant tasks and benchmarks. The long-form generation attempt in the appendix is interesting, and could perhaps be developed more and moved into the main text?
>
> We agree that it would be exciting to see LeReT applied to other datasets. At the time, HotpotQA and HoVeR were the only two multi-hop datasets with document annotations. Tackling long-form generation requires a concerted effort around data generation, which we believe is beyond the scope of this project. We are excited about future work that tackles this direction!

---

### Meta-Review · Area_Chair_MKpG · 2024-12-21

**Metareview:**

This submission introduces LeReT, a reinforcement learning framework that helps LLMS improve their search queries. The motivation is that LLMs often struggle with formulating the right queries. LeReT aims to address this by optimizing query quality through trial and error. The experiments show that the proposed method enhances the retrieval accuracy by up to about 30% and improving downstream evaluations by 17%.

The reviewers identified its strengths as:
- the empirical results suggest that RL improves query generation for multi-hop queries and retrieval accuracy.
- the iterative training proposed brings performance improvement by enhancing retrieval and grounding.
- the proposed framework is compatible with various retrieval systems, making possible applications.

It also received concerns on
- novelty, the reviewers find that the approach seems incremental, with relatively limited novelty compared to existing RAG or QA methods.
- scalability, by relying on direct supervision, limiting scalability and requiring high computational resources for multi-hop retrieval.
- experiments, the experimental setup could be improved by employing latest datasets and comparisons with recent works.

After the rebuttal, reviewers 9WtD and cwbp updated their ratings to 6, and reviewers v1Yv and NW9U unfortunately did not engaged with the authors' rebuttal. Overall this submission receives the borderline ratings of 6, 6, 6, 6. Overall, the proposed method is empirically simple, flexible, and can be applied to any off-the-shelf retriever, making it a promising technique for enhancing LLM pipelines. Of course, as pointed out by the reviewers, the technical novelty does not form as the strength of this work. Given the relatively possible ratings and possible applications, this work at its current format receives an acceptance recommendation.

**Additional Comments On Reviewer Discussion:**

After the rebuttal, reviewers 9WtD and cwbp updated their ratings to 6, and reviewers v1Yv and NW9U unfortunately did not engaged with the authors' rebuttal. Overall this submission receives the borderline ratings of 6, 6, 6, 6.

---

### Decision · Program_Chairs · 2025-01-22

Accept (Poster)